# Titanium Surfaces with a Laser-Produced Microchannel Structure Enhance Pre-Osteoblast Proliferation, Maturation, and Extracellular Mineralization In Vitro

**DOI:** 10.3390/ijms25063388

**Published:** 2024-03-16

**Authors:** Yi-Wen Chen, Tao Chiang, I-Hui Chen, Da-Yo Yuh, Hsiu-Yang Tseng, Chuang-Wei Wang, Hsin-Han Hou

**Affiliations:** 1Department of Dentistry, National Taiwan University Hospital, Taipei 100, Taiwan; evelyn.peri@gmail.com; 2Graduate Institute of Clinical Dentistry, School of Dentistry, National Taiwan University, Taipei 100, Taiwan; 3Biomate Implant Academy Institute, Kaohsiung 806, Taiwan; yunglintc@gmail.com; 4Division of Periodontology, Department of Dentistry, Kaohsiung Medical University Hospital, Kaohsiung 807, Taiwan; 1010391@gap.kmu.edu.tw; 5Division of Periodontology, Department of Dentistry, Tri-Service General Hospital, Taipei 114, Taiwan; yuhdayo@gmail.com; 6Department of Dentistry, National Defense Medical Center, Taipei 114, Taiwan; 7Department of Mechanical Engineering, National Taiwan University of Science and Technology, Taipei 106, Taiwan; htseng@ntust.edu.tw; 8Department of Dermatology, Drug Hypersensitivity Clinical and Research Center, Chang Gung Memorial Hospital, Taoyuan 333, Taiwan; kiruamairo@gmail.com; 9Graduate Institute of Oral Biology, School of Dentistry, National Taiwan University, Taipei 100, Taiwan

**Keywords:** titanium, laser, mesenchymal stem cells, cell proliferation, cell differentiation

## Abstract

The clinical success of dental titanium implants is profoundly linked to implant stability and osseointegration, which comprises pre-osteoblast proliferation, osteogenic differentiation, and extracellular mineralization. Because of the bio-inert nature of titanium, surface processing using subtractive or additive methods enhances osseointegration ability but limits the benefit due to accompanying surface contamination. By contrast, laser processing methods increase the roughness of the implant surface without contamination. However, the effects of laser-mediated distinct surface structures on the osteointegration level of osteoblasts are controversial. The role of a titanium surface with a laser-mediated microchannel structure in pre-osteoblast maturation remains unclear. This study aimed to elucidate the effect of laser-produced microchannels on pre-osteoblast maturation. Pre-osteoblast human embryonic palatal mesenchymal cells were seeded on a titanium plate treated with grinding (G), sandblasting with large grit and acid etching (SLA), or laser irradiation (L) for 3–18 days. The proliferation and morphology of pre-osteoblasts were evaluated using a Trypan Blue dye exclusion test and fluorescence microscopy. The mRNA expression, protein expression, and protein secretion of osteogenic differentiation markers in pre-osteoblasts were evaluated using reverse transcriptase quantitative polymerase chain reaction, a Western blot assay, and a multiplex assay, respectively. The extracellular calcium precipitation of pre-osteoblast was measured using Alizarin red S staining. Compared to G- and SLA-treated titanium surfaces, the laser-produced microchannel surfaces enhanced pre-osteoblast proliferation, the expression/secretion of osteogenic differentiation markers, and extracellular calcium precipitation. Laser-treated titanium implants may enhance the pre-osteoblast maturation process and provide extra benefits in clinical application.

## 1. Introduction

Titanium implants are a common therapeutic choice for whole or partial teeth deficiency [1]. The success of a clinical dental implant is highly associated with implant stability and osseointegration capacity [2]. Osseointegration is defined as the direct connection of surfaces and functional interaction between bone and implant [3]. The osseointegration level is dependent on pre-osteoblast maturation, which comprises pre-osteoblast proliferation, osteogenic differentiation, and extracellular mineralization [4]. In the early stage, the surrounding original bone provides primary mechanical stability. Following the processes of angiogenesis and osseointegration, the newly formed bone provides secondary stability. Between the second and sixth weeks, the decrease in primary mechanical stability is larger than that in secondary stability [5,6]. Stability loss results in a critical situation for dental implants. Therefore, enhancing osseointegration while ensuring implant stability remains a major challenge for titanium implants.

Titanium and titanium alloys have high biocompatibility, corrosion resistance, and biomechanical ability [7]. However, given the bioinert nature of titanium, additional surface-treating techniques are needed to facilitate osseointegration [7]. Previous studies have found that surface treatment with an Nd:YVO4 laser on titanium implants leads to phase and structural changes occurring, as well as a titanium recast layer forming on the implant surface through rapid solidification [8]. Thus, the percentage of bone–implant contact and the rate of bone settlement are affected [9,10]. Collectively, techniques for implant surface processing play a crucial role in delivering optimal osseointegration levels.

Implant manufacturers mostly implement subtractive or additive methods to optimize the surface properties of titanium implants [11,12]. Additive methods include covering the surface of the implant with materials such as titanium paste, hydroxyapatite, calcium phosphate, or bone morphogenetic protein [11]. Subtractive methods include machining, anodizing, sandblasting, large grit acid etching (SLA), and laser irradiation [12]. These surface processing techniques can change the surface energy and wettability of the dental implant and consequently affect the protein adsorption and osseointegration processes [13]. Other studies on micro- and nano-scale surface roughness have reported enhancement of the osseointegration and biomechanical performance of implant materials in vitro and in vivo [14].

Although industrial subtractive or additive surface processes can be beneficial for osseointegration, these techniques risk surface contamination, which can hinder the osseointegration of the implant, representing a major challenge that existing methods cannot circumvent [15]. Hence, laser-treated titanium surfaces have emerged as a promising approach to ameliorate contamination issues. In a previous study [16], modified laser-treatment parameters achieved microchannels averaging 21 µm in width and 18 μm in depth using a 1700-degree Nd:YVO4 source diode-pumped solid-state laser with a wavelength of 355 nm and a pulse duration of 25 ns on ASTM Grade 4 titanium plates. The proliferation, osteogenic differentiation marker, and extracellular mineralization levels of human and mouse stem cells increased when cultured on various energy laser-treated titanium plates with a rough surface [17]. Furthermore, laser-treated titanium plates with microgroove surfaces had a proven affinity with the osteoblast cell line [18]. Controversially, various energy laser-modified titanium plates with a rough surface reduced the viability and activity of human osteoblastic Saos-2 cells [19]. Laser-treated titanium plates exerted no effect on the expression of osteogenic differentiation markers in osteoblast-like cells such as MG-63 cells [20].

Achieving optimal osseointegration levels is crucial to the clinical success of dental implants; laser-treated surfaces enhance the attachment of osteoblasts and reduce contamination risk, unlike conventional additive and subtractive approaches. However, the effects of a novel laser-mediated microgroove structure on pre-osteoblast maturation and osseointegration as well as the underlying mechanisms are not yet fully understood. We hypothesis that laser-mediated microgroove structure promotes osteointegration. Therefore, the aim of this study is to uncover the roles of laser-treated titanium surface in osteointegration.

## 2. Results

Pre-osteoblast human embryonic palatal mesenchymal cells have been proven to differentiate into osteoblasts on titanium plates [21]. In our study, 2.3 × 10^5^ pre-osteoblasts were cultured on titanium plates with G-, SLA-, and L-treated surfaces (Figure 1 and Appendix A) for 3 days. The effects of the three modified titanium surfaces on the growth of pre-osteoblasts were observed and analyzed based on the pre-osteoblast proliferation rate. As shown by the black bars in Figure 2A, the pre-osteoblasts on the L-treated surface grew more quickly. We observed a higher number of cells in the L group than in the G or SLA groups (Figure 2A). These results were also supported by the immunofluorescent images of the EGFP-overexpressed pre-osteoblasts (Figure 2B).

Based on previous studies [4,22,23,24], we examined osteogenic differentiation markers such as COL1A1, DCN, TNFRSF11B, and SPP1 level. According to our qPCR results, the pre-osteoblasts on the L-treated surface significantly increased COL1A1, DCN, TNFRSF11B, and SPP1 mRNA expression (Figure 3). Furthermore, the pre-osteoblasts expressed more COL1A1, DCN, TNFRSF11B, and SPP1 proteins according to the results of the Western blot assay (Figure 4). During osteoblast activation, pre-osteoblasts also secrete several proteins such as COL1A1, OCN, and TNFRSF11B [25,26]. The multiplex assay data showed that pre-osteoblasts on the L-treated surface secreted significantly more COL1, OCN, and TNFRSF11B (Figure 5).

To analyze the mineralization ability in pre-osteoblasts, we initially examined the precipitation of extracellular minerals on the top part of the microchannel that was closed to the nano structure using energy-dispersive X-ray spectroscopy. L-treated titanium plates, when embedded in a noncellular solution, showed no significant difference in calcium or potassium compound precipitation compared to G- or SLA-treated plates (Appendix A). However, when pre-osteoblasts were seeded on the titanium plate, the L group exhibited an increased Alizarin red S stain rate on Day 18, indicating enhanced calcium ion secretion facilitated by the L-treated surface (Figure 6).

## 3. Discussion

There several factors that increase or worsen implant osteointegration, including surgical methods, bone density and volume, post-surgical swelling or infection, smoking behavior, the host’s immune and nutritional condition, and masticatory load [27,28,29,30,31,32,33,34]. Additional considerations include the nature and texture of the implant. The increasing application of titanium in dental implant is due to multiple beneficial effects [35,36]. To overcome titanium bio-inertness in dental implant procedures, subtractive or additive surface modification methods are commonly implemented to create nano- or micro-scale structures and promote osseointegration [37]. However, these surface modification methods have the potential risk of introducing foreign matter to the implant surface during the manufacturing process, thus generating surface contamination, which reduces safety and efficacy. By contrast, laser irradiation is a clean and economical approach with no direct contact with the implant during surface adjustment, ensuring that the product is free from contamination.

Therefore, laser-treated implants have recently started to gain widespread acceptance in the field [38]. Furthermore, laser-treated titanium implants have greater biocompatibility [39] and osseointegration levels [40], with applicability in animals [41,42] and clinical experiments [43]. Although some articles have claimed that the bioactivity of laser-treated titanium surfaces is similar to that of non-laser-treated titanium surfaces [44,45,46], the controversial results may have been caused by variations in experimental parameters, patterns of laser treatment, experimental cell sources, animal models, and clinical study design. Herein, we demonstrated that titanium surfaces with novel laser-mediated microgroove structures increase osseointegration compared with SLA- and G-treated surface groups.

Ideal conditions of an implant surface are produced mainly by generating nano/microporous structures on the implant as a bio-scaffold for functionalization and bio-integration. These nano/microporous surface structures can help to enhance protein adhesion and cell attachment as well as providing anti-biofilm formation ability [47]. A laser-produced microchannel structure on a ZrO_2_ plate has been proven to promote MC3T3-T1 cell proliferation and benefit osseointegration [48]. Laser-mediated titanium plates with similar microchannels have been proven to promote rat calvarial osteoblast cell proliferation and adhesion [16]. Laser-mediated microgrooves on the surface of zirconia promote mouse calvaria-derived pre-osteoblast cell line (MC3T3-E1 cell) adhesion, proliferation, and osteogenic differentiation [49]. Furthermore, a laser-treated titanium surface promotes MG-63 cell adhesion, proliferation, differentiation, and prostaglandin E2 expression compared to those on polished and SLA-treated surfaces [50]. According to our data, the microchannel may play potential roles and may be vital in the promotion of osteointegration ability. A previous study indicated the laser micro-textured titanium-coated silicon surfaces promote human osteosarcoma (HOS) cells’ spreading and proliferation rates [51]. The ~9–11 μm depth has the best combination of contact guidance and HOS cell integration [51]. In our study, the laser-treated titanium surface formed a micro-scale microchannel with a width of 21 µm and a depth of 18 µm on average. Our laser-treated titanium surface has the nano-scaled structure on the top of the microchannel and the relatively smoother surface on the bottom of the microchannel. The human fetal oral fibroblast promotes cellular adhesion in our laser-treated implant with filipodia structure. Although the structure of the laser-generated microchannel is similar to that produced by other conventional techniques, the more complex nanoscale structure around the microchannel in our laser-treated titanium surface provided more climbing sites for pre-osteoblast stretching, which may promote osteogenic gene expression such as parathyroid hormone-related protein gene expression in osteoblasts [52]. Furthermore, the deeper microchannels in our laser-treated titanium surface may create more space that facilitates nutrient supplements for pre-osteoblasts [53]. However, additional experiments are required for verification in clinical applications.

The results of both mRNA and protein levels of COL1, SPP1, DCN, and TNFRSF11B were significantly increased by 3-day culturing of pre-osteoblasts on the L-treated titanium plate. The pre-osteoblast maturation process involves proliferation, osteogenic differentiation, and extracellular mineralization [4,21,22,54], each governed by specific osteogenic differentiation markers that regulate cellular responses. For example, COL1A1 is upregulated during osteoblast differentiation and SPP1 is a marker for extracellular matrix maturation. DCN modulates collagen matrix assembly and mineralization [55] and regulates the cell cycle [56], which is expressed during extracellular matrix mineralization. In addition, TNFRSF11B gene-translated osteoprotegerin is a suppressor for bone resorption [57]. Physiological stretching of the cell influences the expression of insulin-like growth factor and mechanogrowth factor [58]. A previous study indicated that mechanical stretching promotes Runx2 expression via the JAK2/STAT3 pathways through mechanosensor polycystin-1 [59] in human osteoblastic cells. Upregulated COL1, SPP1, DCN, and TNFRSF11B in human embryonic palatal mesenchymal cells cultured on laser-generated microchannels may result from the action of mechanical stretch force. Exploring the involved mechanosensory, downstream molecules and transcriptional factors can further strengthen the beneficial effects of laser-treated titanium implants.

In this study, we only confirm the effects of laser-treated surfaces on HEPM activity. The validation in other cells, such as rat bone marrow mesenchymal stem cells (rBMSCs) and MC3T3-E1 cells, will further confirm the role of laser-treated surfaces in osteointegration. The in vitro study may limit the overall evaluation of osteointegration processes. We will apply the implant with a laser-treated surface on the rat tibia to assay the effects on the osseointegration level in vivo. Herein, we can deeply explore the detailed mechanism and involved molecules that participate in the regulation of osteointegration. In a stage earlier than osteointegration, the mesenchymal fibroblast-like cell and endothelial play crucial roles in angiogenesis during implant implantation immediately [36]. After the stable stage, the antibacterial ability in processing implants can help to eliminate the possibility of peri-implantitis [60]. In the future, we will further evaluate the effects of titanium implants with laser-treated surfaces on angiogenesis and antibacterial ability.

## 4. Materials and Methods

### 4.1. Cell Culture

Pre-osteoblast human embryonic palatal mesenchymal cells (CRL-1486TM, ATCC, Manassas, VA, USA) were cultured and placed in a 37 °C incubator with 5% CO_2_ in Dulbecco’s modified Eagle’s medium/F12 (1:1) (21041-025, Gibco Invitrogen, Carlsbad, CA, USA) supplemented with 10% fetal bovine serum and 1% antibiotic Penicillin-Streptomycin-Amphotericin B solution [21].

### 4.2. Titanium Plate and Implant

A titanium plate (Grade 4, diameter: 15 mm; thickness: 1 mm) was used for the in vitro study. Laser-treated titanium plates were purchased from Biomate (Taipei, Taiwan), with laser treatment as previously described [16]. The laser-treated surfaces were produced through controlled micro-ablation using a pulsed Nd:YVO_4_ source diode-pumped solid-state laser (355 nm wavelength) with Q-Switch output. Emission in Q-Switch mode allowed the generation of brief impulses (tenths of a nanosecond), which removed the material from the surface. This operation mode ensured that the process would have high repeatability without thermally altering surrounding areas, avoiding cluster formation and any contamination of the surface by elements other than titanium [53]. The laser beam power was set to generate parallel microgrooves with a width of 21 µm and a pitch of 18 µm. Laser-treated specimens were also subjected to cleaning treatment after laser micro texturing. To assess the beneficial effects of the laser (L) treatment, we compared it with grinding (G) and SLA surface treatments. The G treatment used silicon carbide water sandpaper at #1200 grit. For the SLA titanium plate group, Al_2_O_3_ was used for sandblasting, and the plate was then placed in a strong acid solution for acid etching. All the plates were sterilized before use. After surface treatment, the plates were subjected to a field-emission scanning electron microscope test to determine the deposition of calcium and phosphorus compounds on the plate surface (on the top part of the microchannel and closed to the nanostructure).

### 4.3. Cell Number Determination

The pre-osteoblasts were incubated on G-, SLA-, and L-treated plates for 3 days and then transferred to new plates. A 0.4% Trypan Blu dye exclusion test was conducted to measure the number of cells in the new and old plates and compare them to determine the growth rate of cells on the three titanium plates.

### 4.4. Cell Morphology

Images of the enhanced green fluorescent protein (EGFP)-overexpressed pre-osteoblasts were taken using an optical/confocal fluorescence microscope (BA410E series biological microscope, Motic, NJ, USA) to ultimately evaluate the effects of the processed-titanium surfaces on pre-osteoblast morphology.

### 4.5. Reverse Transcription Quantitative Polymerase Chain Reaction

The pre-osteoblasts were incubated on G-, SLA-, and L-treated plates for 3 days. Based on previous studies [22,23,24], we selected four osteogenic differentiation markers, collagen type I alpha 1 chain (COL1A1), decorin (DCN), TNFRSF11B, and osteospomtin (SPP1), for evaluation. The treated cells were subjected to TRIzol reagent (15596026, Invitrogen) for mRNA extraction. Subsequently, 100 ng of purified cellular mRNA was used for the reverse transcription reaction using the Power SYBR™ Green RNA-to-CT™ 1-Step Kit. The corresponding cDNA was used for a quantitative polymerase chain reaction (qPCR) assay in a Biorad CFX96 (Bio-Rad, Hercules, CA, USA) system with primer sets of COL1A1, DCN, TNFRSF11B, SPP1, and GAPDH (Table 1).

### 4.6. Protein Extraction and Western Blot Assay

After 3 days of incubation on G-, SLA-, and L-treated plates, pre-osteoblasts were lysed using radioimmunoprecipitation assay buffer (W-7849-500, Goal Bio, Taipei, Taiwan). Cellular lysates were centrifuged at 12,000 rpm for 5 min for supernatant removal. The extracted protein was quantified using a protein assay and equal amounts of protein were separated using 10% sodium dodecyl-sulfate polyacrylamide gel electrophoresis and transferred to an Amersham Hybond P 0.45 µm polyvinylidene difluoride membrane (10600023, GE Healthcare, Chicago, IL, USA). After being blocked with 5% skimmed milk, the membranes were incubated with various primary antibodies and then incubated with the corresponding secondary antibodies. The protein bands were detected using an Amersham ECL Select Western Blotting Detection Reagent (RPN2235, GE Healthcare) and quantified using Image Quant 5.2 software (Healthcare Bio-Sciences, Pittsburgh, PA, USA).

### 4.7. Multiplex Assay

The pre-osteoblasts were incubated on G-, SLA-, and L-treated plates for 3 days. The medium was then collected to be applied in a multiplex assay with Procarta Plex Multiplex Immunoassay Kits and Human ProcartaPlex Panels (Thermo Fisher, Waltham, MA, USA). In accordance with previous studies [25,26], we selected four osseointegration markers, collagen I (COL I), osteocalcin (OCN), TNFRSF11B, and SPP1 for evaluation.

### 4.8. Extracellular Mineralization Assay

The pre-osteoblasts were incubated on G-, SLA-, and L-treated plates for 4 and 18 days. After staining with Alizarin red S, images were captured using an optical microscope. The deposition of calcium ions during mineralization from the extracellular matrix was evaluated (red-brown calcium compound staining). The red-brown calcium compounds were dissolved with 10% (*v*/*v*) acetic acid and quantified using a spectrophotometer at optical density (OD540).

### 4.9. Statistical Analysis

For comparisons among data from more than two experimental conditions, we first used an analysis of variance to check for alterations, followed by a post hoc Tukey’s test for qPCR and Western blotting to identify significant differences between two specific groups. A *p*-value less than 0.05 was considered statistically significant.

## 5. Conclusions

Laser-treated titanium surfaces significantly enhanced pre-osteoblast proliferation and promoted COL1, SPP1, DCN, and TNFRSF11B mRNA and protein expression/secretion, as well as extracellular mineralization. These findings strongly suggest that a laser-treated titanium surface with a microgroove structure has profound beneficial effects on pre-osteoblast maturation and osseointegration and holds great potential for clinical applications.

## Figures and Tables

**Figure 1 ijms-25-03388-f001:**
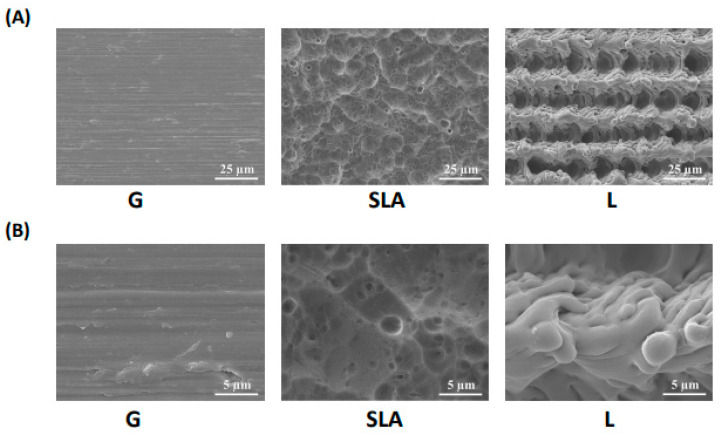
Surface morphology of titanium implant-treated surfaces. Scanning electron microscope images of titanium plates with G-, SLA-, and L-treated surfaces. The scale bar is 25 μm for the upper column (**A**) and 5 μm for the lower column (**B**).

**Figure 2 ijms-25-03388-f002:**
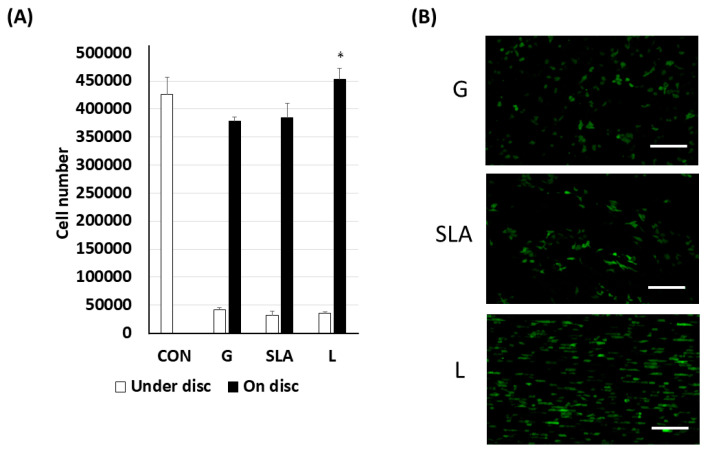
Enhanced pre-osteoblast proliferation on L-treated titanium surfaces. The 2.3 × 10^5^ pre-osteoblast cells (**A**) or enhanced green fluorescent protein-overexpressed pre-osteoblast cells were grown on/under the titanium plates with G-, SLA-, and L-treated surfaces. The cell number was determined using a Trypan Blu dye exclusion test. (**B**) Visualization of EGFP-overexpressed pre-osteoblast proliferation on titanium plates with G-, SLA-, and L-treated surfaces. A scale bar of 200 μm is used for the fluorescent imaging. * *p* < 0.05 vs. G group.

**Figure 3 ijms-25-03388-f003:**
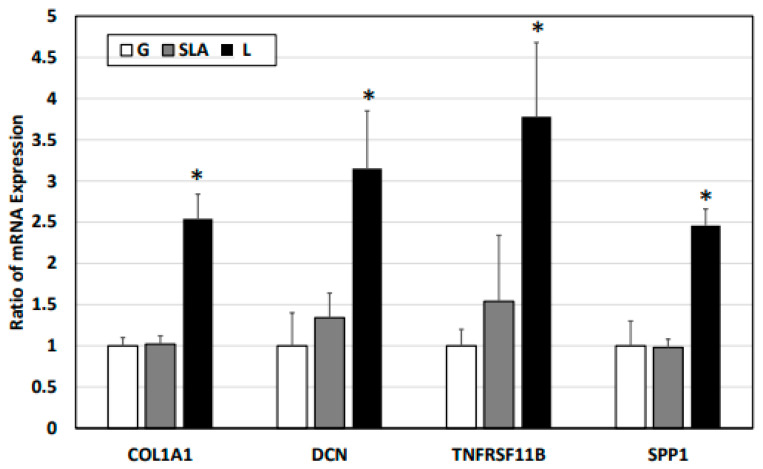
L-treated titanium surface upregulation of the expression of osteogenic differentiation markers in pre-osteoblasts obtained via a qPCR assay. The 2.3 × 10^5^ pre-osteoblast cells were incubated on titanium plates with G-, SLA-, and L-treated surfaces for 72 h. Purified cellular mRNA was used for a qPCR assay with primer sets of COL1A1, DCN, TNFRSF11B, and SPP1. * *p* < 0.05 vs. G group.

**Figure 4 ijms-25-03388-f004:**
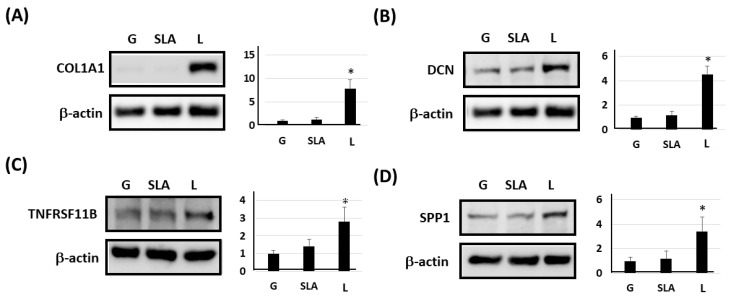
L-treated titanium surface upregulation of the expression of osteogenic differentiation markers in pre-osteoblasts obtained using a Western blot assay. The 2.3 × 10^5^ pre-osteoblast cells were incubated on titanium plates with G-, SLA-, and L-treated surfaces for 72 h. Purified cellular protein lysate was used for a Western blot assay with antibodies of (**A**) COL1A1, (**B**) DCN, (**C**) TNFRSF11B, (**D**) SPP1, and β-actin. * *p* < 0.05 vs. G group.

**Figure 5 ijms-25-03388-f005:**
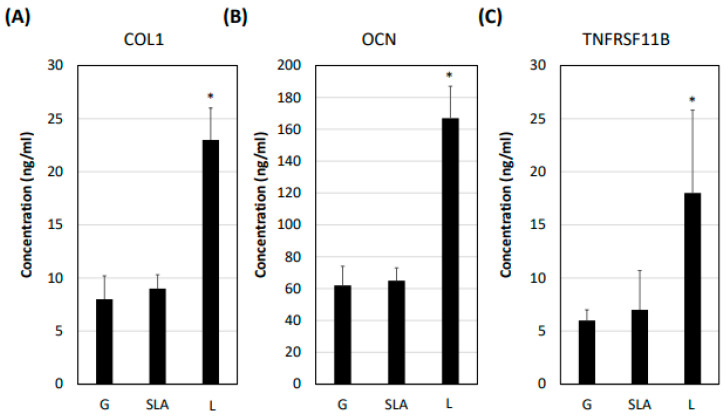
L-treated titanium surface promotion of osteogenic differentiation marker secretion in pre-osteoblasts. The 2.3 × 10^5^ pre-osteoblast cells were incubated on titanium plates with G-, SLA-, and L-treated surfaces for 72 h. The cellular medium was collected and applied in a multiplex assay to determine the concentration of (**A**) COL1A1, (**B**) OCN, and (**C**) TNFRSF11B. * *p* < 0.05 vs. G group.

**Figure 6 ijms-25-03388-f006:**
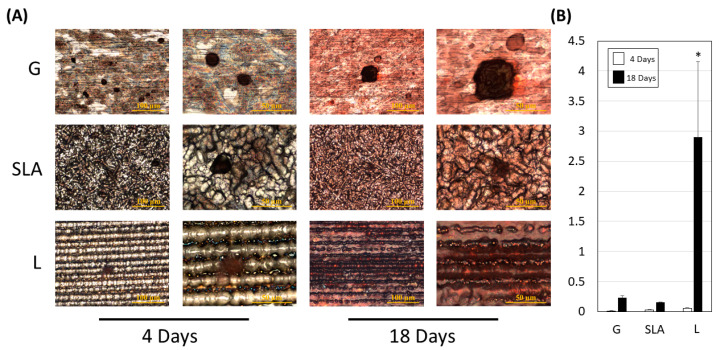
L-treated titanium surface promotion of pre-osteoblast extracellular mineralization. (**A**) Images were collected after a 4- or 18-day culture of 2.3 × 10^5^ pre-osteoblast cells on titanium plates with G-, SLA-, and L-treated surfaces. A scale bar of 100 μm is shown in the left column and a scale bar of 50 μm is shown in the right column. The reddish-brown calcium compound indicates the mineralization ability of pre-osteoblasts. (**B**) Quantitative results are shown according to the OD540 level. A scale bar of 100 μm and 50 μm is used for the imaging in left and right column respectively. * *p* < 0.05 vs. G group.

**Table 1 ijms-25-03388-t001:** The primer set of genes used in qPCR.

Gene Name	Forward	Reverse
*COL1A1*	GATTCCCTGGACCTAAAGGTGC	AGCCTCTCCATCTTTGCCAGCA
*DCN*	AGCTGAAGGAATTGCCAGAA	CTCTGCTGATTTTGTTGCCA
*TNFRSF11B*	GAACCCCAGAGCGAAATAC	CGCTGTTTTCACAGAGGTC
*SPP1*	CGAGGTGATAGTGTGGTTTATGG	GCACCATTCAACTCCTCGCTTTC
*GAPDH*	GTCTCCTCTGACTTCAACAGCG	ACCACCCTGTTGCTGTAGCCAA

## Data Availability

Data is contained within the article and Appendix A. The data presented in this study are available on request from the corresponding author.

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
