# Peer review of "Titanium Surfaces with a Laser-Produced Microchannel Structure Enhance Pre-Osteoblast Proliferation, Maturation, and Extracellular Mineralization In Vitro"

_ijms, 2024, doi:10.3390/ijms25063388_

Round 1

Reviewer 1 Report

Comments and Suggestions for Authors

Interesting work on the osteoproliferative capacity of implant surfaces treated with superficial laser. Just some criticisms listed below:

-The abstract section (lines 21-27) appears too redundant, I ask you to summarize it

-Check that all keywords are Pubmed MESH terms

-At the end of the introduction section, insert the objectives of the study and the null hypotheses which will be refuted at the end of the work in light of the results obtained

-Remove the primer sets from the text and make an appropriate table

-In the discussion section, after an initial sentence on the problem, insert a sentence that summarizes the results obtained in the work.

-Some considerations on the factors that increase or worsen implant osteointegration should be added. In this regard, speaking of masticatory load, I ask you to include the following scientific work in the reference section:

Grande F, Pozzan MC, Marconato R, Mollica F, Catapano S. Evaluation of Load Distribution in a Mandibular Model with Four Implants Depending on the Number of Prosthetic Screws Used for OT-Bridge System: A Finite Element Analysis (FEA). Materials (Basel). 2022 Nov 10;15(22):7963. doi: 10.3390/ma15227963. PMID: 36431449; PMCID: PMC9699052.

-Another aspect worth mentioning is the different behavior under load of the implants compared to dental elements, live or devitalized. In this regard, I ask you to include the following scientific work in the reference section:

Chieruzzi M, Rallini M, Pagano S, Eramo S, D'Errico P, Torre L, Kenny JM. Mechanical effect of static loading on endodontically treated teeth restored with fiber-reinforced posts. J Biomed Mater Res B Appl Biomater. 2014 Feb;102(2):384-94. doi: 10.1002/jbm.b.33017. Epub 2013 Sep 2. PMID: 24000235.

-A section on the limitations of the study is missing

Author Response

Interesting work on the osteoproliferative capacity of implant surfaces treated with superficial laser. Just some criticisms listed below:

-The abstract section (lines 21-27) appears too redundant, I ask you to summarize it

Thanks for your crucial suggestion, we have rewritten the paragraph and the edited version is at lines 21-26.

-Check that all keywords are Pubmed MESH terms

Thanks for your crucial suggestion, we checked the keyword on the website (https://www.ncbi.nlm.nih.gov/mesh) and the edited version is at line 43.

-At the end of the introduction section, insert the objectives of the study and the null hypotheses which will be refuted at the end of the work in light of the results obtained

Thanks for your crucial suggestion, we have rewritten the paragraph and the edited version is at lines 96-99.

-Remove the primer sets from the text and make an appropriate table

Thanks for your crucial suggestion, we have rewritten the paragraph and the edited version is at lines 145-148 and Table 1.

-In the discussion section, after an initial sentence on the problem, insert a sentence that summarizes the results obtained in the work.

Thanks for your crucial suggestion, we have rewritten the paragraph and the edited version is at lines 262-264, 282-285 and 294-296.

-Some considerations on the factors that increase or worsen implant osteointegration should be added. In this regard, speaking of masticatory load, I ask you to include the following scientific work in the reference section:

Grande F, Pozzan MC, Marconato R, Mollica F, Catapano S. Evaluation of Load Distribution in a Mandibular Model with Four Implants Depending on the Number of Prosthetic Screws Used for OT-Bridge System: A Finite Element Analysis (FEA). Materials (Basel). 2022 Nov 10;15(22):7963. doi: 10.3390/ma15227963. PMID: 36431449; PMCID: PMC9699052.

-Another aspect worth mentioning is the different behavior under load of the implants compared to dental elements, live or devitalized. In this regard, I ask you to include the following scientific work in the reference section:

Chieruzzi M, Rallini M, Pagano S, Eramo S, D'Errico P, Torre L, Kenny JM. Mechanical effect of static loading on endodontically treated teeth restored with fiber-reinforced posts. J Biomed Mater Res B Appl Biomater. 2014 Feb;102(2):384-94. doi: 10.1002/jbm.b.33017. Epub 2013 Sep 2. PMID: 24000235.

Thanks for your crucial suggestion, we have rewritten the paragraph and the edited version is at lines 243-247 and added reference 35.

-A section on the limitations of the study is missing

Thanks for your crucial suggestion, we have added the paragraph and the edited version is at lines 311-322.

Reviewer 2 Report

Comments and Suggestions for Authors

The communication entitled ‘Titanium surfaces with a laser-produced microchannel structure enhance pre-osteoblast proliferation, maturation, and extracellular mineralization in vitro’ reports the pre-osteoblast proliferation, the expression/secretion of osteogenic differentiation markers, and extracellular calcium precipitation on laser treated titanium surfaces. The following points are recommended to improve the manuscript.

Line 108 -  titanium plate grade shall be provided

Line 123-124 – From where,  the deposition of calcium and phosphorus compounds is expected to occur. Clearly mention this.

Mention the make and model of equipment used for characterisation.

It is suggested to provide the primer sequence in a separate table

Visualization of pre-osteoblast proliferation on titanium plates – image quality is not up to the standard.

It is recommended to improve the discussion section by adding details about how the laser modification can be correlated with the osteogenic differentiation markers, instead of detailing about published work more focus shall be given on the observations from the present study.

Author Response

Reviewer 2

The communication entitled ‘Titanium surfaces with a laser-produced microchannel structure enhance pre-osteoblast proliferation, maturation, and extracellular mineralization in vitro’ reports the pre-osteoblast proliferation, the expression/secretion of osteogenic differentiation markers, and extracellular calcium precipitation on laser treated titanium surfaces. The following points are recommended to improve the manuscript.

Line 108 - titanium plate grade shall be provided

Thanks for your crucial suggestion, we have rewritten the paragraph and the edited version is at line 108.

Line 123-124 – From where, the deposition of calcium and phosphorus compounds is expected to occur. Clearly mention this.

Thanks for your crucial suggestion, we have rewritten the paragraph and the edited version is at lines 125-126 and 228-229.

Mention the make and model of equipment used for characterisation.

Thanks for your crucial suggestion, we have added the paragraph and the edited version is at lines 134-135 and 163-164.

It is suggested to provide the primer sequence in a separate table

Thanks for your crucial suggestion, we have rewritten the paragraph and the edited version is at lines 145-148 and Table 1.

Visualization of pre-osteoblast proliferation on titanium plates – image quality is not up to the standard.

The original figure was copied from a PowerPoint slide. To improve the quality of the figure, we translated Figure 2 into a PDF file and attached in revised version.

It is recommended to improve the discussion section by adding details about how the laser modification can be correlated with the osteogenic differentiation markers, instead of detailing about published work more focus shall be given on the observations from the present study.

Thanks for your crucial suggestion, we have added the paragraph and the edited version is at lines 274-285.

Reviewer 3 Report

Comments and Suggestions for Authors

Comments to the authors

The objective of this study is to understand how laser-produced microchannels on titanium surface affects pre-osteoblast maturation, which is quite important topic for the filed of dentistry. In this manuscript, the authors successfully concluded the potential benefit of laser-produced microchannels method, which improved pre-osteoblasts survival, differentiation and mineralization. The manuscript was well written and all data were solid enough. However, it seemed not to be clearly described the detail of the laser-produced microchannels methos and the difference from previous methods. Several concerns which may need to be addressed before the publication are listed below;

Specific comments

 1. It might not be clearly described about the novel points (neues) of this study in the introduction section. It would be better to emphasize the uniqueness of this laser-produced microchannels method compared with previous reports using similar methods.  

2. It would be better to be described the detail about this laser-produced microchannels method by adding several simple schemas or pictures of actual material in the materials and methods section.

3. The last paragraph of the introduction (Line 94-99) may be better to be moved to the discussion section, because this paragraph may mislead readers to expect in vivo data. Also, if the authors would have in vivo data showing the stability of the laser-produced microchannels treated dental implant in alveolar bone, it would be wonderful to add.

4. Are human palate cells the best for this experiment? Although this reviewer understood this cell line was used in previous study, did the authors have any alternative options of cell line for this study? It would be better to clarify that part in the manuscript.

5. About Fig.2B, what is the meaning of the GFP positive cells? Did the authors perform any staining or just autofluorescence? Why didn’t the authors perform co-staining with DAPI?

6. About Fig.2, the authors concluded the proliferation was enhanced. However, the data just showed the increase of cell numbers in L group. This might be because of the enhanced proliferation as the authors described, or could be because of the suppression of cell death. It would be required to examine cell proliferation using Ki67 antibody or something. Also, what would be the potential mechanisms regulating cell proliferation, differentiation, and mineralization by the changes of titanium surface? Specific cellular signaling pathway? Cell cycle? Or something else?

Author Response

R3

Comments to the authors

The objective of this study is to understand how laser-produced microchannels on titanium surface affects pre-osteoblast maturation, which is quite important topic for the filed of dentistry. In this manuscript, the authors successfully concluded the potential benefit of laser-produced microchannels method, which improved pre-osteoblasts survival, differentiation and mineralization. The manuscript was well written and all data were solid enough. However, it seemed not to be clearly described the detail of the laser-produced microchannels methos and the difference from previous methods. Several concerns which may need to be addressed before the publication are listed below;

Specific comments

  1. It might not be clearly described about the novel points (neues) of this study in the introduction section. It would be better to emphasize the uniqueness of this laser-produced microchannels method compared with previous reports using similar methods.

Thanks for your crucial suggestion, we have rewritten the paragraph and the edited version is at lines 92-99 in the introduction section and lines 265-292.

  1. It would be better to be described the detail about this laser-produced microchannels method by adding several simple schemas or pictures of actual material in the materials and methods section.

Thanks for your crucial suggestion, we have described the details of titanium plate preparation (lines 107-126). The actual material was provided in supplemental figure 2.

  1. The last paragraph of the introduction (Line 94-99) may be better to be moved to the discussion section, because this paragraph may mislead readers to expect in vivo data. Also, if the authors would have in vivo data showing the stability of the laser-produced microchannels treated dental implant in alveolar bone, it would be wonderful to add.

Sorry for the error typo, it should be “in vitro” instead of “in vivo”. To correct this, we rewrote the paragraph and the edited version is at lines 96-99.

  1. Are human palate cells the best for this experiment? Although this reviewer understood this cell line was used in previous study, did the authors have any alternative options of cell line for this study? It would be better to clarify that part in the manuscript.

Thanks for your crucial suggestion. In the future, the results could be validated in mouse calvaria-derived preosteoblast cell line or rat bone marrow mesenchymal stem cells (rBMSCs), MC3T3-E1 (ATCC; Manassas, VA) to confirm the findings in this study. We have rewritten the paragraph and the edited version is on lines 311-314.

  1. About Fig.2B, what is the meaning of the GFP positive cells? Did the authors perform any staining or just autofluorescence? Why didn’t the authors perform co-staining with DAPI?

The cell was EGFP-overexpressed HEPM and we performed an immunofluorescent stain of cytoskeleton markers and DAPI in the next study. We have rewritten the paragraph and the edited version is on lines 133, 187 and 198.

  1. About Fig.2, the authors concluded the proliferation was enhanced. However, the data just showed the increase of cell numbers in L group. This might be because of the enhanced proliferation as the authors described, or could be because of the suppression of cell death. It would be required to examine cell proliferation using Ki67 antibody or something. Also, what would be the potential mechanisms regulating cell proliferation, differentiation, and mineralization by the changes of titanium surface? Specific cellular signaling pathway? Cell cycle? Or something else?

Thanks for your crucial suggestion. We seeded 2.3*10^5 cells on a plate and after 72 hours, the cell numbers were about 4.2, 3.7, 3.75 and 4.5*10^5 (sorry for the mistype on cell number in original figure 2) in the vehicle, G, SLA and L group respectively. Therefore, the cell number was increased in all groups. Therefore, we conclude laser-treated surface increase the HEPM proliferation. We speculate the stretching modification in HEPM on laser-changed titanium structure mediated the results in this study. However, the potential mechanisms (including specific gene expression, cellular signaling pathway, cell cycle change or others) involved in laser-treated surface-promoted cell proliferation, differentiation, and mineralization remain worth exploring. We have rewritten the paragraph and the edited version is at lines 314-317 and Figure 2.
